organic chemistry/supramolecular chemistry

$Hg^{2+}$ adsorbent, chitosan, calix[4]arene, urea cross-link, kinetic, isotherm

**Authors for correspondence:**
Reza Zadmard
e-mail: zadmard@ccerci.ac.ir
Gholamreza Ahmadian
e-mail: ahmadian@nigeb.ac.ir

# Synthesis of a new chitosan-*p-tert*-butylcalix[4]arene polymer as adsorbent for toxic mercury ion

Fahimeh Hokmabadi[1], Reza Zadmard[1], Ali Akbarzadeh[1], Vida Tafakori[2], Mohammad Reza Jalali[1] and Gholamreza Ahmadian[3]

[1]Chemistry and Chemical Engineering Research Center of Iran, Tehran, Iran
[2]Department of Cell and Molecular Biology, Faculty of Science, Kharazmi University, Tehran, Iran
[3]Department of Industrial and Environmental Biotechnology, National Institute of Genetic Engineering and Biotechnology (NIGEB), Pajoohesh BLVD, Tehran, Iran

RZ, 0000-0001-7563-4911

In this paper, we have synthesized a novel chitosan-*p-tert*-butylcalix[4]arene polymer (CCP) as a highly efficient adsorbent for mercury ion ($Hg^{2+}$) removal from water. In fact, a lower rim diamine derivative of *p-tert*-butylcalix[4]arene has been cross-linked with chitosan chain by carbonyl diimidazole (CDI) as the linker. CDI forms a urea linkage between calix[4]arene diamine derivative and amine groups of the chitosan polymeric chain. The structure and properties of the new polymer were characterized by Fourier transform infrared spectroscopy, X-ray diffraction and scanning electron microscope. Also, the adsorption capacity of CCP was studied towards $Hg^{2+}$ in aqueous medium by inductively coupled plasma-optical emission spectrometry. Interestingly, the results showed a considerable adsorption capacity for CCP in comparison with chitosan. Therefore, CCP can be introduced as a promising adsorbent for the elimination of $Hg^{2+}$ from wastewaters. Moreover, because of the conformity of adsorption kinetic with pseudo-second-order kinetic models, it can be concluded that chemical adsorption has an important role between functional groups on CCP polymer and $Hg^{2+}$ ions. In addition, according to Freundlich isotherm, the CCP surface was heterogeneous with different functional groups.

## 1. Introduction

Today, water pollution by industrial and urban wastewaters has become a major concern owing to its adverse effects on human

and wildlife health [1]. Among these pollutants, toxic heavy metal ions such as mercury ion ($Hg^{2+}$) have been classified as priority hazardous substances [2]. In particular, $Hg^{2+}$ even at very low concentrations is highly toxic and causes serious damage to proteins and enzymes through covalent binding to their active thiols, called S-mercuration [3,4]. So far, various methods have been developed to remove toxic heavy metal ions from wastewaters including reverse osmosis [5], electrochemical treatment [6], coagulation [7], bio-removal approaches [8], ion-exchange membrane [9], chemical precipitation [10] and adsorption [11]. The use of adsorbents is the most economical and effective method, owing to its high efficiency (especially at a low concentration of Hg), selectivity, simplicity and low cost [12]. Adsorbents are generally made of porous organic or inorganic materials such as polymers [13], silica [14], alumina [15,16] and activated carbon [17,18] which have been modified with other appropriate materials [19]. Chitosan is a nitrogenous polysaccharide which is obtained from deacetylation of chitin as the most abundant bio-polymer after natural cellulosic carbohydrate polymers [20–22]. Owing to its outstanding features like non-toxicity [23], permeability [24], metal complexation [25], biocompatibility [26] and good film-forming properties [27], chitosan has been used in a wide range of applications such as surgical sutures [28], drug delivery [29], artificial skin [30], cosmetics [31], wastewaters treatment [32] and adsorbents [33]. Many specific properties of chitosan such as hygroscopicity and metal complexation lie in the presence of hydroxyl, amino and N-acetyl groups over the polymeric chain [34]. Although the ability of chitosan to bind to metal ions makes chitosan an ideal candidate for the removal of metal cations from waste solutions, high solubility of chitosan in acidic solvents might be considered as a serious drawback to its use as a metal adsorbent in various pH conditions [35,36]. So, chemical modifications of chitosan could be a reasonable approach to improve the properties of chitosan as a superb adsorbent [25,36,37]. Calix[4]arenes are basket-like cyclic phenol formaldehyde tetramers that have been widely used in supramolecular chemistry. In fact, they are regarded as the third-generation host molecules, following crown ethers and cyclodextrins owing to their unique conical architecture [38,39]. The calixarene structure includes a polar (lower) rim, a non-polar (upper) rim and a hydrophobic cavity. The phenolic hydroxyls in the polar rim and the aromatic moiety in the non-polar rim can be easily functionalized by the conventional synthetic transformations to achieve desirable supramolecular systems [40]. As a result of flexible conformation and ease of structural and functional modifications, the internal cavity of calixarenes could be a promising versatile host for different chemical species like ions and organic molecules [41–43]. The combination of chitosan with calixarene scaffolds provides a great possibility to improve different properties of chitosan such as solubility, selectivity and adsorption capacity towards metal ions. The reason for this is the synergic action between chitosan and calix[n]arene moieties [44]. There are several studies in which various calixarene–chitosan polymers have been introduced for adsorption of different chemicals such as heavy metal cations ($Co^{2+}$, $Hg^{2+}$, $Cu^{2+}$, $Cd^{2+}$, $Pb^{2+}$, $Ni^{2+}$ and $Cr_2O_7$) [45], di-n-butyl phthalate [46] and dyes [47]. Here, we report a new method for the immobilization of calix[4]arene onto the chitosan polymer. The new calixarene–chitosan polymer has been characterized and evaluated for $Hg^{2+}$ ions adsorption in the aqueous medium. Furthermore, in order to determination of the adsorption kinetic and isotherm, the effects of the contacting time and the initial metal ion concentration were studied.

# 2. Material and methods

## 2.1. Materials

All chemicals and solvents used in this research were purchased from Merck Millipore, Germany. Chitosan was purchased from Sigma Aldrich Co., USA. The degree of deacetylation was approximately 75% and the average molecular weight was medium. All of the materials were used without further purification.

## 2.2. Instrumentation

Melting points were determined with a Buchi B-545 apparatus. Proton nuclear magnetic resonance ($^1H$ NMR) spectra were recorded on a Bruker-ARX500 spectrometer. Fourier transform infrared (FTIR) spectrum was recorded on a Bruker, Vector 22 infrared spectrometer. The morphology of the polymers, after the surfaces were coated with gold, was investigated by scanning electron microscopy (SEM) (TESCAN, Vega3); X-ray diffraction (XRD) studies were performed on a Bruker, D8 advance XRD instrument. Inductively coupled plasma adsorptions were recorded on a Spectro Arcos inductively coupled plasma-optical emission spectrometry spectrometer.

**Scheme 1.** Synthesis route of 1,3-diamine derivative of *p-tert*-butylcalix[4]arenen precursor **4**.

## 2.3. Synthesis of chitosan-*p-tert*-butylcalix[4]arene polymer

Calixdiamine **4** has been synthesized in a good yield, using a previously described procedure [48] (scheme 1). To a suspension of 0.2 g (0.8 mmol) of chitosan in 5 ml of water containing 1% (v/v) acetic acid, 0.12 g (0.8 mmol) of carbonyl diimidazole (CDI) was added. The mixture was stirred under nitrogen atmosphere at room temperature for 2 h. Then, 0.32 g (0.4 mmol) of calixdiamine **4** in 10 ml of dichloromethane (DCM) containing 5 ml of dimethylformamide (DMF) was added to the mixture and heated at 70°C for 20 h (scheme 2). After cooling to room temperature, the precipitate was filtrated and washed with 50 ml of methanol and 50 ml of DCM, and dried under vacuum at 70°C to give 0.4 g of desired product as a light-yellow powder: yield 65%; m.p. (decomposition): 370°C; FTIR (KBr, $v$, cm$^{-1}$): 3369, 2958, 1671, 1480, 1201, 1051, 873, 751.

## 2.4. Evaluation of chitosan-*p-tert*-butylcalix[4]arene polymer as mercury adsorbent

The adsorption experiments were performed to evaluate the adsorption of Hg$^{2+}$ ion by chitosan-*p-tert*-butylcalix[4]arene polymer (CCP). To a 25 ml of aqueous solution of Hg$^{2+}$ (as chloride salt) at a concentration of 125 nmol ml$^{-1}$ (pH 7, 25°C), 25 mg of CCP was added. Subsequently, the mixture was stirred for 90 min at 25°C and then filtered. The residual metal ion in aqueous phase was measured by inductively coupled plasma (ICP) spectroscopy. To calculate the percentage of the metal adsorbed by the new CCP, the following formula was used:

$$E(\%) = \frac{C_i - C_f}{C_i} \times 100, \tag{2.1}$$

here, $E$ refers to the adsorption percentage of polymer for the metal cation; $C_i$ and $C_f$ refer to the metal cation concentrations before and after adsorption, respectively. All experiments were carried out twice to calculate the average amounts of adsorbed cations. In these experiments, row chitosan was used as a control.

## 2.5. Mercury adsorption studies

To examine kinetic models, equilibrium isotherms and the effect of different variables on Hg$^{2+}$ adsorption, more adsorption experiments were conducted. To this end, a solution of 1000 nmol ml$^{-1}$ Hg$^{2+}$ was prepared by dissolving 2.70 g of analytical grade HgCl$_2$ in 100 ml of distilled water. The solution was stored as a stock solution. Other concentrations of HgCl$_2$ solution (10, 25, 75, 100 and 125 nmol ml$^{-1}$) were prepared by diluting the stock solution. The mixtures of CCP and Hg$^{2+}$ solutions were mixed at 25°C and 700 r.p.m. for 1.5 h and then filtered. The residual Hg$^{2+}$ ion concentration

**Scheme 2.** Synthesis route of chitosan-*p-tert*-butylcalix[4]arene (CCP).

($C_f$) in filtrated samples were measured in aqueous phase using ICP spectroscopy. In addition, control experiments were performed using raw chitosan. To calculate the percentage of $Hg^{2+}$ adsorption by the CCP, equation (2.1) was used.

## 2.6. Adsorption kinetics of mercury ion on chitosan-*p-tert*-butylcalix[4]arene polymer

Considerable information on the reaction pathways, the mechanisms of the adsorption reaction, and solute uptake is obtained from the adsorption kinetics data. To explain the kinetic of adsorption, the pseudo-first-order and pseudo-second-order adsorption models were used [49]. According to the Pseudo-first-order model, the rate of adsorption site occupation is relative to the number of unoccupied sites [50]. The linear equation for the pseudo-first-order model is

$$\text{Log}\,(q_e - q_t) = \text{Log}\,q_e - \left(\frac{K_1}{2.303}\right)t, \tag{2.2}$$

here, $q_e$ and $q_t$ refer to the amounts of metal ions adsorbed at equilibrium and at any time ($t$), respectively, (nmol mg$^{-1}$) onto the adsorbent surface, and $K_1$ refers to the rate constant of the first-order adsorption [50].

It is assumed that in the pseudo-second-order model, the occupation rate of adsorption sites is relative to the square of the number of unoccupied sites. The linear equation for this model is

$$\frac{t}{q_t} = \left(\frac{1}{K_2 \cdot q_e}\right) + \left(\frac{1}{q_e}\right)t, \tag{2.3}$$

where $q_e$ and $q_t$ refer to the amounts of metal ions adsorbed on the adsorbent at equilibrium and at any time ($t$), respectively, (nmol mg$^{-1}$), and $K_2$ refers to the rate constant of second-order adsorption (mg nmol$^{-1}$ min$^{-1}$) [50].

For determination of kinetic adsorption at 125 nmol ml$^{-1}$ concentrations ($C_i$) in pH 7, 25 mg of CCP was added to 25 ml of an aqueous solution of $Hg^{2+}$. Then, the mixtures were stirred for 15, 30, 45, 60, 75 and 90 min at 25°C and then filtered. The $Hg^{2+}$ concentration ($C_f$) was assessed using ICP spectroscopy. In addition, a control experiment was performed using raw chitosan. The linear regression curve-fitting procedure was carried out using Microsoft Excel. To examine the goodness of fit of the data to the model, the coefficient of determination, $R^2$ and least-squares method were used.

## 2.7. Adsorption isotherms of mercury ion on chitosan-*p-tert*-butylcalix[4]arene polymer

In general, equilibrium adsorption isotherms can be used to determine the capacity, surface properties and adsorption affinity. In these theoretical models, the Langmuir and Freundlich equilibrium models are more common. The linear Langmuir equation is

$$\frac{C_e}{q_e} = \left(\frac{C_e}{q_{max}}\right) + \left(\frac{1}{K_L \cdot q_{max}}\right), \tag{2.4}$$

here $q_e$ refers to the equilibrium adsorption capacity of adsorbent in nmol $Hg^{2+}$/mg adsorbent, $C_e$ refers to the equilibrium concentration of $Hg^{2+}$ ion in nmol $ml^{-1}$, $q_{max}$ refers to the maximum amount of $Hg^{2+}$ adsorbed in nmol $mg^{-1}$ of adsorbent, and $K_L$ refers to the constant value that is known as the adsorption bonding energy in nmol $ml^{-1}$ [51].

Langmuir isotherm indicates a homogeneous adsorption, where each molecule has a constant enthalpy and adsorption activation energy (all sites possess equal affinity for the adsorbate), with no transmigration of the adsorbate in the plane of the surface [52].

The linear Freundlich equation is

$$\text{Log } q_e = \text{Log } K_f + 1/n \text{ Log } C_e,  \tag{2.5}$$

here, $q_e$ refers to the equilibrium adsorption capacity of the adsorbent in nmol $Hg^{2+}$ /mg adsorbent, $C_e$ refers to the equilibrium concentration of $Hg^{2+}$ ion in nmol $ml^{-1}$, and $K_f$ (in nmol $ml^{-1}$) and $1/n$ refer to the constants related to the adsorption capacity and intensity, respectively [51].

Freundlich isotherm is the first relationship introduced to describe non-ideal and reversible adsorptions, which is not limited to the formation of a monolayer. This empirical model can be used for multi-layer adsorption featured with non-uniform distribution of adsorption heat and affinity over the heterogeneous surface [52].

Isotherm experiments were performed at 25°C, pH 7 according to the procedure mentioned earlier using 25 mg adsorbent for 25 ml of varying initial $Hg^{2+}$ concentrations of 10–125 nmol $Hg^{2+}$ /ml with constant shaking in equilibrium time of 1.5 h. the linear regression fitting was carried out.

The average relative error (ARE) was calculated based on the following equation:

$$\text{ARE} = \frac{100}{n} \sum_{i=1}^{n} \left| \frac{q_{e.meas} - q_{e.calc}}{q_{e.meas}} \right|_i.  \tag{2.6}$$

# 3. Results and discussion

## 3.1. Synthesis of chitosan-*p-tert*-butylcalix[4]arene

Chitosan is a linear polymer with both hydroxyl and amino groups. Here, we activated free amino groups of chitosan by using CDI as the linker to afford compound **6**. Then, 1,3-diamine derivative of *p-tert*-butylcalix[4]arene was added to the reaction mixture. Finally, the mixture was heated to obtain CCP. The final product was characterized by FTIR, XRD and SEM analysis.

## 3.2. Characterization of chitosan-*p-tert*-butylcalix[4]arene polymer

CCP was successfully prepared via the formation of urea linkage between $NH_2$ in chitosan and $NH_2$ in calix[4]arene derivative **4** as a light-yellow powder. CCP was decomposed in temperatures above 370°C. Also, the solubility of CCP has been investigated in several organic solvents such as DCM, dimethyl sulfoxide (DMSO) and DMF. As the results revealed, CCP was practically insoluble in organic solvents and just swollen in acetic acid solution.

## 3.3. Infrared spectral analysis

Figure 1 shows the infrared (IR) transmission spectrums of calixdiamine **4**, raw chitosan and CCP. Based on the obtained IR spectrums, it is obvious that calix[4]arene derivative **4** was covalently immobilized onto the chitosan polymeric chain owing to the appearance of a sharp band at $1671 \text{ cm}^{-1}$ corresponding to the urea carbonyl stretching and the complete disappearance of $NH_2$ bending vibrations at $1593 \text{ cm}^{-1}$ and $1559 \text{ cm}^{-1}$ related to calix[4]arene derivative **4** and chitosan, respectively. Additionally, FTIR spectrum obtained from CCP showed the characteristic peaks of aromatic backbone vibration at $1480 \text{ cm}^{-1}$ (C=C aromatic stretching) and $873 \text{ cm}^{-1}$ (C-H aromatic out-of-plane) because of the presence of calix[4]arene phenyl groups which were not observed in raw chitosan IR spectrum. It was also observed that the OH groups adsorption appeared at $3550-3200 \text{ cm}^{-1}$ and was much stronger in CCP in comparison with calix[4]arene diamine **4** and chitosan. Finally, the peaks at $3000-2850 \text{ cm}^{-1}$ in CCP were much stronger than those in raw chitosan. This observation was attributed to the increase in C-H groups of CCP compared to the chitosan polymeric chain.

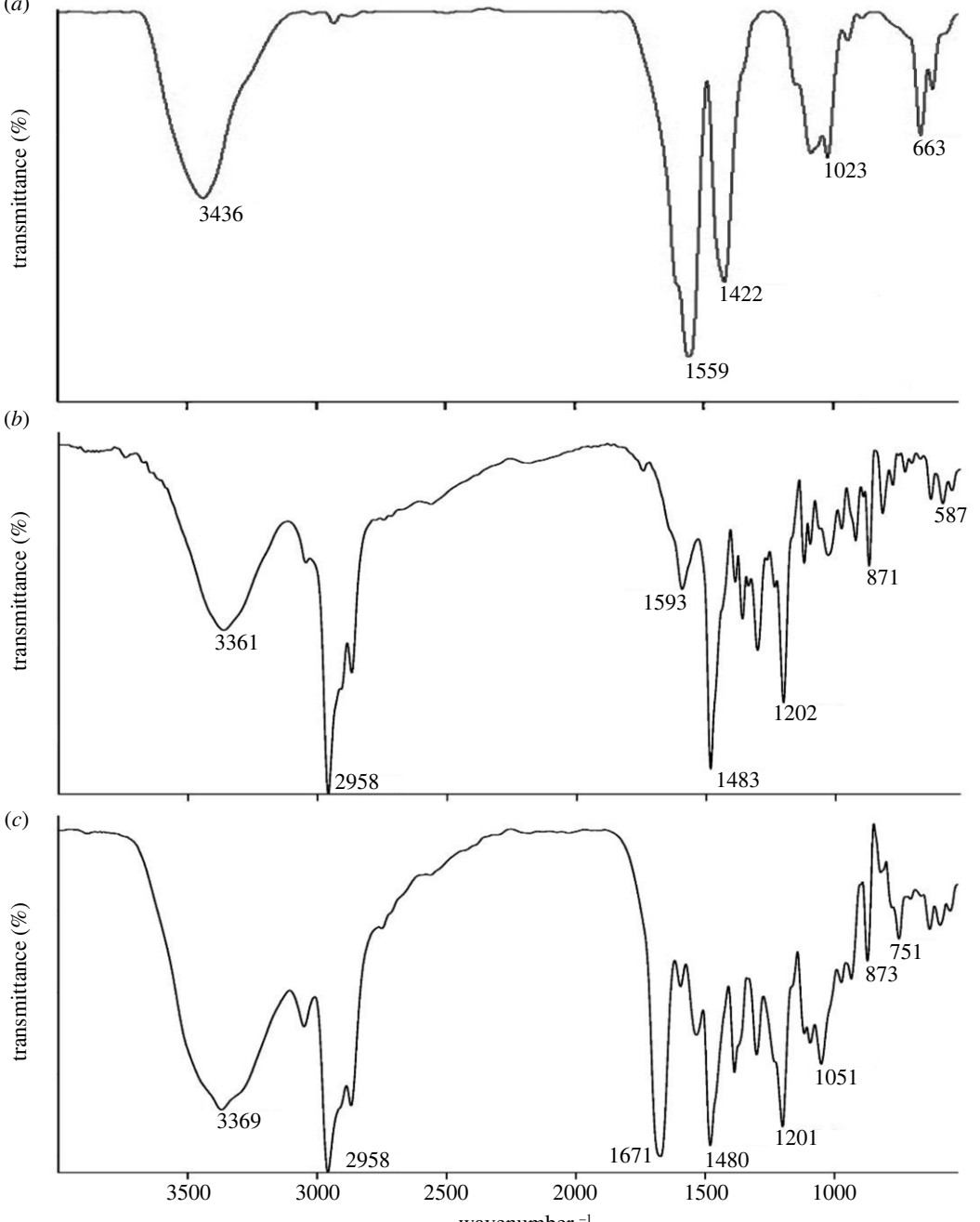

**Figure 1.** The FTIR spectra of (*a*) chitosan, (*b*) compound **4**, and (*c*) CCP.

## 3.4. X-ray diffraction analysis

The XRD patterns of chitosan and CCP are shown in figure 2. Raw chitosan had two characteristic strong wide diffraction peaks at $2\theta = 9°$ and $26°$ which were related to hydrogen bonds and free hydroxyl groups, respectively. However, for CCP, the peak at $2\theta = 9°$ almost disappeared and the peak at $2\theta = 26°$ became weak. It could be attributed to the hydrogen bond deformation in the chitosan backbone, owing to the fact that $NH_2$ groups on the polymeric chain have been involved in the urea linkages formed between chitosan and calixdiamine **4**. The calixarene did not appear in the powdered XRD spectrum owing to their amorphous nature.

## 3.5. Scanning electron microscope characterization

The SEM is an analytical method to produce a magnified image for analysis by scanning the sample with a focused beam of electrons. This technique is used very effectively in microanalysis and failure analysis

(a)

(b)

**Figure 2.** The XRD patterns of (a) chitosan, and (b) CCP.

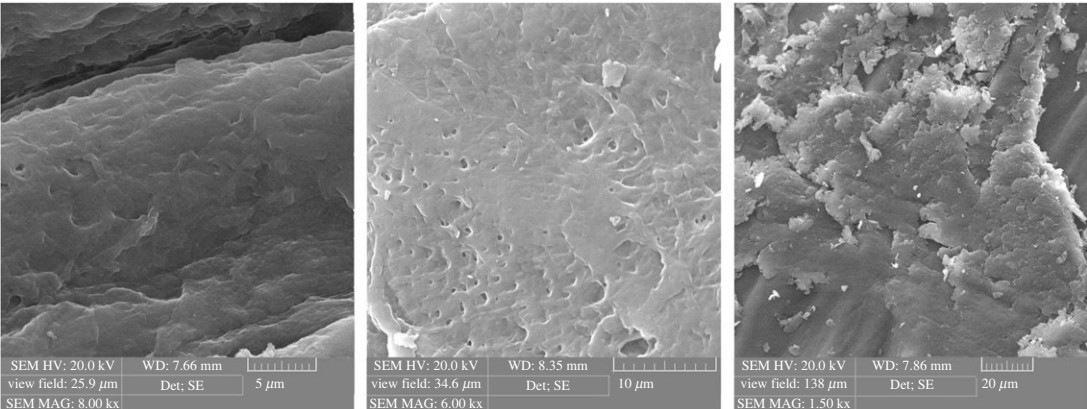

**Figure 3.** The SEM image of chitosan.

of solid inorganic materials [53]. SEMs of the surface morphology of raw chitosan and CCP are presented in figures 3 and 4, respectively. The chitosan has a smooth surface morphology, whereas CCP was significantly different on the surface morphology. The introduction of calixarene units onto chitosan could be the origin of this phenomenon through the destruction of the strong intramolecular

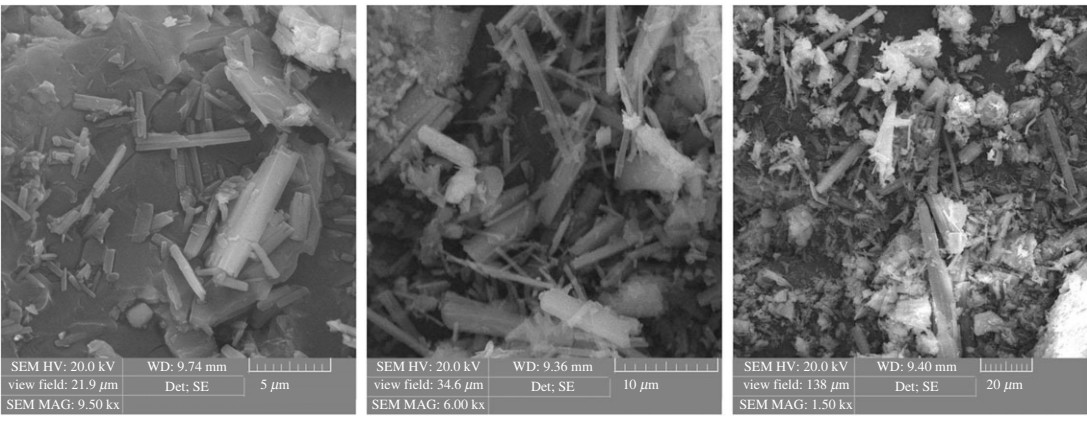

**Figure 4.** The SEM image of CCP.

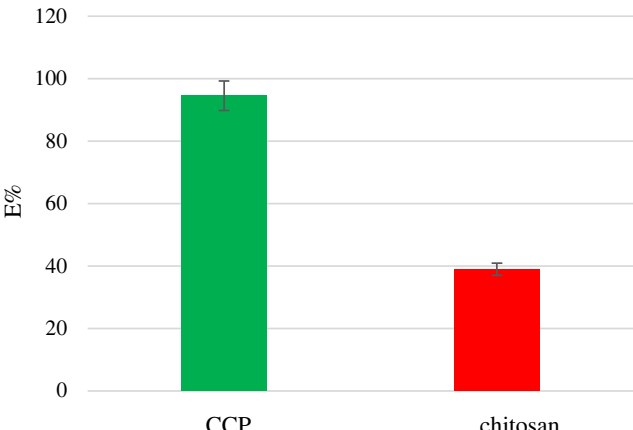

**Figure 5.** Adsorption percentage of Hg$^{2+}$ by CCP compared to raw chitosan. (error bar was added).

hydrogen bonding. This type of morphology is favourable for the preparation of a promising adsorbent with excellent binding ability to heavy metal ions.

## 3.6. Evaluation of chitosan-*p-tert*-butylcalix[4]arene polymer as an adsorbent

According to the obtained results, CCP showed a significant adsorption capacity for Hg$^{2+}$ (94.6%) compared to raw chitosan (39%) (figure 5). It is possible that calixarene aromatic cavity attached to chitosan provide many electron-rich sites on the polymeric chain which can establish more interactions with Hg$^{2+}$ ions related to raw chitosan.

Until now, several chitosan-based adsorbents for Hg$^{2+}$ ion have been reported in the literature. Table 1 summarizes the maximum capacity and contacting time of some similar adsorbents to compare with this work.

## 3.7. Effect of contacting time

The effect of contacting time (0–90 min) on the adsorption of Hg$^{2+}$ by CCP is illustrated in figure 6. The equilibrium time was found to be 60 min. At equilibrium conditions, 94.6% of Hg$^{2+}$ was removed from aqueous solution. In these experiments, raw chitosan used as a control.

## 3.8. Adsorption kinetic

Kinetic studies were carried out to examine both Hg$^{2+}$ adsorption rate and adsorption efficiency on CCP. To illustrate the reaction mechanism of Hg$^{2+}$ adsorption by CCP, the experimental findings were interpreted using pseudo-first- and pseudo-second-order kinetic models. As it can be seen in figure 7,

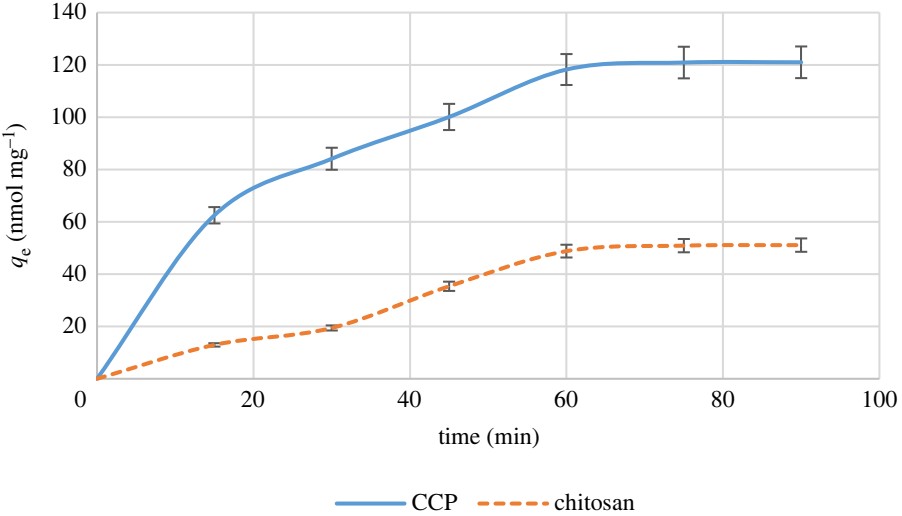

**Figure 6.** Effect of contacting time on the adsorption capacity.

**Table 1.** Comparison of various chitosan-based adsorbents for $Hg^{2+}$ with CCP.

| no. | adsorbent | contacting time | adsorption (%) | reference |
|---|---|---|---|---|
| 1 | calix[4]arene-cross-linked chitosans (CTS–CA) | 12 h | 63.8 | [20] |
| 2 | calix[4]arene-based chitosan polymer (C[4]BCP) | 1 h | 93.3 | [45] |
| 3 | calix[4]arene–chitosan polymer (C[4]CSP) | 1 h | 89.46% | [21] |
| 4 | calix[6]arene–chitosan polymer (C[6]CSP) | 1 h | 98.96% | [21] |
| 5 | calix[8]arene–chitosan polymer (C[8]CSP) | 1 h | 97.85% | [21] |
| 6 | gelatin–chitosan hydrogel particles | 30 min | 82.00% | [54] |
| 7 | green nanocomposite based on chitosan and Brassica Gongylodes leaf extract. | 45 min | 87.19% | [55] |
| 8 | CCP | 1 h | 94.60% | present work |

in the pseudo-first-order kinetic model, the correlation coefficient value or $R^2$ was calculated to be 0.9497. According to the formula obtained from the diagram, $y = -0.0287x + 1.756$, log $q_e = 1.756$ and, $q_e = 57.01$ nmol mg$^{-1}$, but practically the value of $q_e$ was found to be 69.49 nmol mg$^{-1}$. The difference between experimental and theoretical values of $q_e$ is also owing to the inconsistency of this model with the results obtained from the study. In other words, the occupancy rate of adsorption sites is not proportional to the number of unoccupied sites. The value of the rate constant $K_1$ is 0.0287 min$^{-1}$, which indicates that the acceleration of the uptake of adsorption sites based on this model, is slow. This is against our results, which in the first 60 min, the surface adsorption reaches the equilibrium state.

In order to find a model that was more in coincidence with the practical results, a second-order kinetic model was used and the $t/q_t$ versus $t$ graph was plotted. As shown in figure 8, the correlation coefficient value $R^2$ was calculated to be 0.993. Therefore, it can be concluded that the results are highly consistent with the pseudo-second-order kinetic model. This fact implies the existence of chemical adsorption between the adsorbent and the adsorbed metals and the metal adsorption process follows this model.

According to the equation obtained from the diagram ($y = 0.015x + 0.026$), the value of the rate constant $K_2$ was found 0.0086 g mg$^{-1}$ min$^{-1}$. Meanwhile, the calculated value of $q_e$ from this equation was 66.66 nmol mg$^{-1}$, which is very close to its practical value of 69.49 nmol mg$^{-1}$, and this is another good reason that the practical data obtained is consistent with this model.

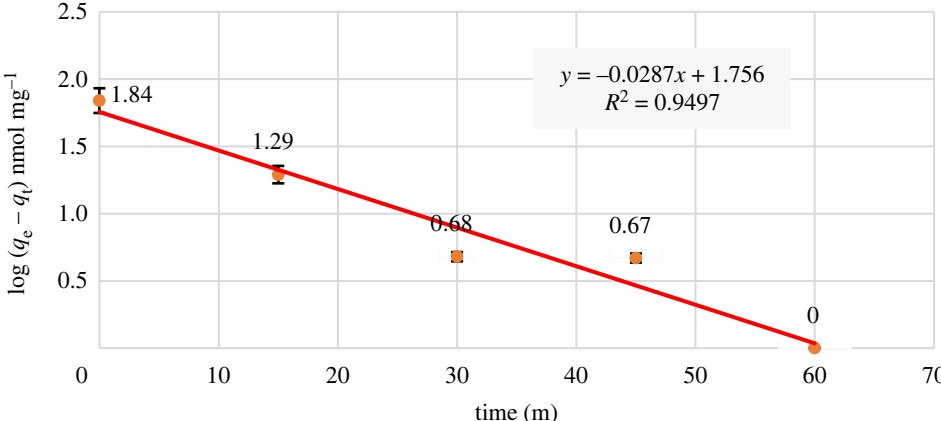

**Figure 7.** Pseudo-first-order kinetics graph.

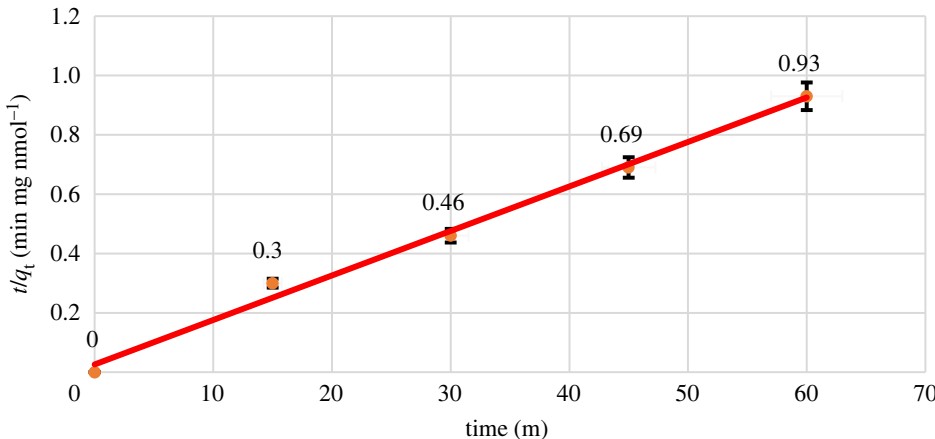

**Figure 8.** Pseudo-second-order kinetics graph.

**Table 2.** The parameters for kinetic models.

|  | pseudo-first order | pseudo-second order |
|---|---|---|
| $k_i$ | 0.0287 | 0.0086 |
| $q_e$ | 57.0164 | 66.6666 |
| ARE | 0.2151 | 0.0628 |
| $R^2$ | 0.9497 | 0.9932 |

Following the pseudo-second-order kinetic model implies the existence of chemical adsorption between the adsorbent and the adsorbed metals and shows that the rate of uptake of adsorption sites is proportional to the square of the amount of unoccupied sites [51].

Obtained parameters from the both kinetic models as well as the ARE, are summarized in table 2. As expected the ARE value for the pseudo-second-order model is less than the first-order model.

## 3.9. Effect of mercury ion concentration on adsorption process

In the process of discontinuous adsorption, the initial concentration of metal ions plays an important role as the force overcoming the mass transfer resistance between the solution and the solid adsorbent surface [49]. Therefore, it is expected that with increasing concentration of metal ion in the solution, the amount of the adsorption will increase. As figure 9 shows, the amount of $Hg^{2+}$ adsorption on CCP enhanced as the concentration of $Hg^{2+}$ in the solution was raised.

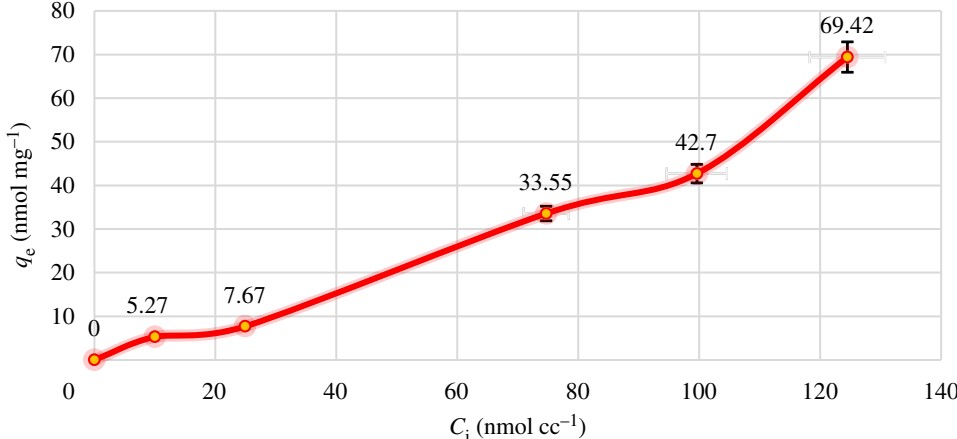

**Figure 9.** Effect of $Hg^{2+}$ concentrations on the adsorption capacity.

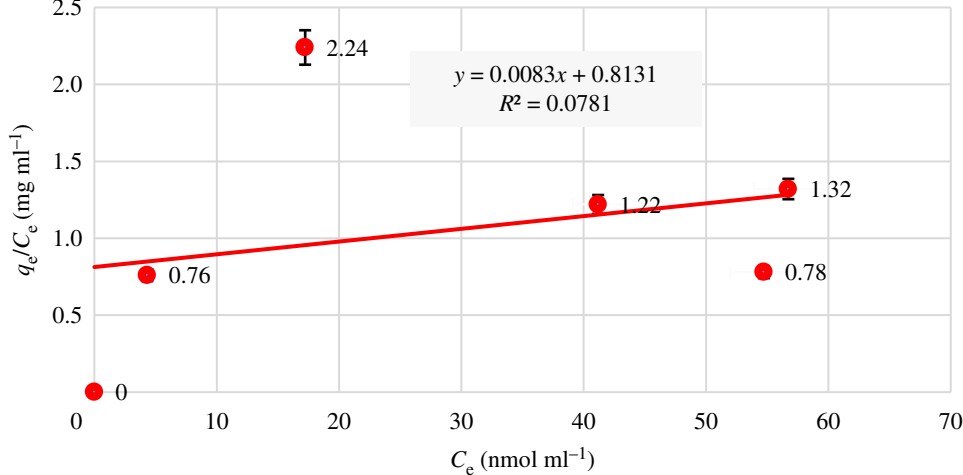

**Figure 10.** Langmuir adsorption isotherm.

## 3.10. Adsorption isotherm

The chemistry of the adsorption process is relatively complex. The two most common models to study these processes are the Langmuir and Freundlich models. In order to evaluate the adsorption performance of the metal ions on the adsorbent surface, the compliance of practical information with the Langmuir isotherm was studied. For this purpose, a diagram of $C_e/q_e$ versus $C_e$ (equilibrium concentration of $Hg^{2+}$) was drawn (figure 10). According to the correlation coefficient value, $R^2 = 0.07$, the adsorption process does not match the Langmuir model. So, it can be concluded that the adsorption does not occur as a single layer.

Therefore, in order to investigate the compliance of the process of $Hg^{2+}$ adsorption on the adsorbent surface, the Freundlich model was investigated. For this purpose, the log $q_e$ diagram was drawn against log $C_e$.

As can be seen in figure 11, the $R^2$ value is 0.94, which means that the experimental information is more coincidental with the Freundlich model. Moreover, the ARE for the Freundlich isotherm model is calculated as 0.2421 which is significantly less than that for the Langmuir isotherm model (0.2825). According to this model, there are different levels of energy at the adsorption sites, which show that the adsorption process occurs in multi-layers. This means different functional groups were involved in the interaction between CCP and $Hg^{2+}$. These findings are in conformity with the kinetic study results. The value of $1/n$ is equal to 1.13, which indicates that there is chemical adsorption (not physical) between the adsorbent and the metal ions. This finding is in coincidence with the information obtained from the kinetic analysis of the practical data and introduces a pseudo-second-

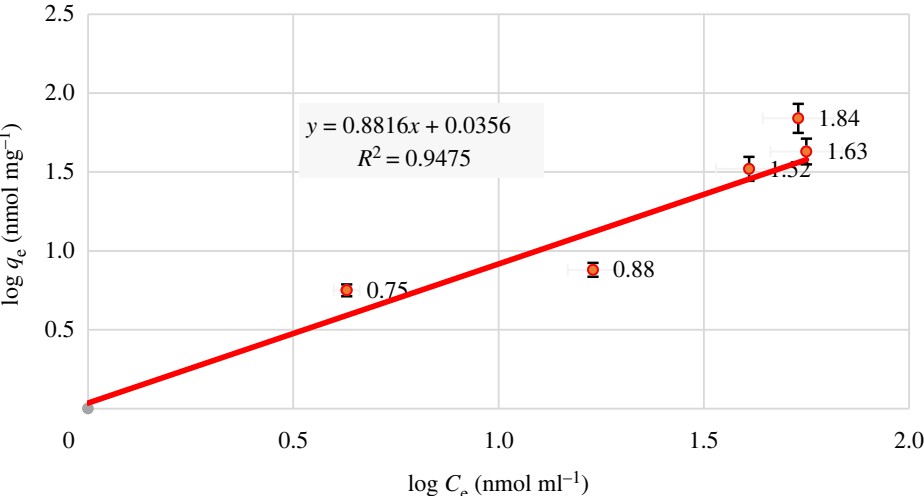

**Figure 11.** Freundlich adsorption isotherm.

order model for the adsorption process. Furthermore, the adsorption capacity can be obtained from the diagram as $K_f = 1.08$ nmol mg$^{-1}$ for the dry weight of the adsorbent.

## 4. Conclusion

In conclusion, CCP was synthesized as a new CCP through the formation of urea linkage between chitosan and the lower rim diamine derivative of *p-tert*-butylcalix[4]arene **4** via a simple procedure in good yield. The cross-link formation between the chitosan polymeric chain and calixdiamine **4** via CDI was confirmed by FTIR spectroscopy. Furthermore, the adsorption capacity of CCP was investigated towards Hg$^{2+}$ in aqueous medium. The results showed excellent adsorption properties for Hg$^{2+}$ in comparison with chitosan. In fact, 25 mg of CCP could extract 94% of metal ions from aqueous solution of Hg$^{2+}$ in 60 min which was threefold higher than raw chitosan. Additionally, CCP is stable in acidic conditions and can easily be powdered. Therefore, CCP can outperform raw chitosan in term of elimination of Hg$^{2+}$ from wastewaters. These advantages lie in the presence of calixarene moieties on the chitosan polymeric chain which provides several aromatic electron-rich sites on the polymer and changes its morphology.

Data accessibility. All data are available in the manuscript or uploaded as the electronic supplementary material [56].
Authors' contributions. F.H.: conceptualization, data curation, formal analysis and methodology; R.Z.: conceptualization, data curation, formal analysis, investigation, methodology and project administration; A.A.: visualization and writing—original draft; V.T.: formal analysis, software and validation; M.R.J.: investigation, supervision and visualization; G.A.: conceptualization, investigation, project administration and writing—original draft.
    All authors gave final approval for publication and agreed to be held accountable for the work performed therein.
Conflict of interest declaration. The authors declare that they have no known competing financial interests.
Funding. This work was supported by the Chemistry and Chemical Engineering Research Center of Iran.

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
