## [Peer Review File · Royal Society Open Science]

Review History

RSOS-211223.R0 (Original submission)

Review form: Reviewer 1

Is the manuscript scientifically sound in its present form?

Yes

Are the interpretations and conclusions justified by the results?

Yes

Is the language acceptable?

Yes

Do you have any ethical concerns with this paper?

No

Have you any concerns about statistical analyses in this paper?

Yes

Recommendation?

Major revision is needed (please make suggestions in comments)

Comments to the Author(s)

Comments

The research article "Synthesis of a new chitosan-p-tert-butylcalix[4]arene polymer as adsorbent for toxic Hg²⁺ metal ion" seems to be a good work by the researchers, however it requires a major revision so as to be published in this reputed journal.

1. The introduction section needs improvement in terms of language and content. The authors should consult native English speaker and rewrite the introduction and rewrite the introduction with the updated references.
 - a. RSC Advances 2019, 10 (5), 2943-2943
 - b. ACS omega 2019, 4 (17), 17425-17437
 - c. International journal of biological macromolecules, 2019, 136, 189-198
2. The authors have focussed mainly on synthesis and characterization of the product, however the adsorption properties which is the application part needs attention.
3. Effect of pH on the adsorption sites and the adsorption of Hg²⁺ on chitosan-p-tert-butylcalix[4]arene polymer must be explained with point of zero charge of the adsorbent.
4. The selective adsorption of Hg²⁺ on chitosan-p-tert-butylcalix[4]arene polymer must be justified by adsorption kinetics and thermodynamic data. Furthermore, the spontaneity of the adsorption of Hg²⁺ on chitosan-p-tert-butylcalix[4]arene polymer must be explained using Van't Hoff plots
5. Error bars must be incorporate with all the data plots, to justify the selective adsorption of of Hg²⁺ on chitosan-p-tert-butylcalix[4]arene polymer.
6. Mere calculation of adsorption data from the kinetic and isotherm plots is not enough for conclusion of the adsorption phenomena. The validation of the results must be ensured by statistical error analysis by the sum of the square of the error (SSE), sum of absolute error (SAE), chi square (χ^2), and standard deviation Δq (%), between the experimental data and calculated values.

Review form: Reviewer 2 (Tugba Isik)

Is the manuscript scientifically sound in its present form?

No

Are the interpretations and conclusions justified by the results?

Yes

Is the language acceptable?

Yes

Do you have any ethical concerns with this paper?

No

Have you any concerns about statistical analyses in this paper?

No

Recommendation?

Accept with minor revision (please list in comments)

Comments to the Author(s)

I would like to thank to the authors for their detailed research about this topic. This manuscript employs the adsorption of Hg²⁺ ions from water by using functionalized chitosan. The authors compared the adsorption capacity of newly synthesized functionalized-chitosan with the ones without functional group. Then, they characterized the structural properties through FTIR, XRD and SEM. The Hg⁺ adsorption capacity of synthesized material was investigated by ICP-MS. The research is quite comprehensive and includes enough physical characterization techniques for the synthesized polymers. Also, the authors studied the adsorption kinetics by Freundlich and Langmuir isotherms. According to the presented results, they reported that 25 mg of synthesized polymer could adsorb 94% of metal ions from 25 mL of 125 nmol/ml aqueous Hg²⁺ solution. The research contributes to the current literature about the utilization of functionalized-chitosan for metal ion removal applications.

I would recommend that this manuscript is suitable for publication in 'Royal Society of Open Science' after the consideration of suggested points.

You can find my comments and suggestions below.

- The language of the manuscript should be reviewed. There are repetitive conjunctions in same sentences.
- The authors should add the details about pH during the adsorption and they should include a pH study that shows the adsorption of Hg²⁺ vs pH of solution.
- There are some other studies which use chitosan for the metal ion removal. They might be cited such as Biomacromolecules 2010, 11, 12, 3301–3308; RSC Adv., 2013,3, 7828–7837.
- In Figure 1, the peaks might be labelled to show the functional groups.
- In section 3.5, the paragraph starts with 'SEM also known as SEM microscopy'. I think, it's not a true definition because SEM abbreviation already includes 'microscopy' term.
- In Figure 3 and 4, there are (a), (b), (c) sections that identify the magnification of micrographs. The authors gave the scale bar dimensions in the caption but it seems like the size of chitosan. If they can give the magnification values, the caption will be more clear.
- In Figure 6, the standard deviation of chitosan was given horizontally. Is it true? I think standard deviations give the deviation from average adsorption (q_e) value but it seems like the deviation from time.
- In Section 3.9, the last sentence says that 'This means there are two kind of adsorption process between different functional groups of the CCP and Hg²⁺.' What do you mean here? Why does it depend on the concentration? Could you give more details about it?
- The results are given as % sorption by using X mg of polymer from X mL of X mol solution. I think it is pretty confusing. The discussion about the adsorption results should be done as mg (ion) per g (polymer) (mg/g) – the units may change. It makes the result comparable with the current literature.
- In the last part, there is a need for discussion about the results. The literature studies about the Hg²⁺ removal by using chitosan should be compared with this research's results. Why these polymers are promising and why people should prefer this method?
- What about the performance of polymers in the presence of competitive double valent ions? Do the authors have any result about selectivity of polymer?
- What is the reusability property of polymers? Can we regenerate them and how is the adsorption / desorption cycle?

Review form: Reviewer 3

Is the manuscript scientifically sound in its present form?

Yes

Are the interpretations and conclusions justified by the results?

Yes

Is the language acceptable?

Yes

Do you have any ethical concerns with this paper?

No

Have you any concerns about statistical analyses in this paper?

No

Recommendation?

Accept with minor revision (please list in comments)

Comments to the Author(s)

Equation 2 is missing t, please amend.

Decision letter (RSOS-211223.R0)

Dear Dr Zadnard:

Title: Synthesis of a new chitosan-p-tert-butylcalix[4]arene polymer as adsorbent for toxic Hg²⁺ metal ion

Manuscript ID: RSOS-211223

The editor assigned to your manuscript has now received comments from reviewers. We would like you to revise your paper in accordance with the referee and Subject Editor suggestions which can be found below (not including confidential reports to the Editor). Please note this decision does not guarantee eventual acceptance.

Please submit your revised paper before 06-Feb-2022. Please note that the revision deadline will expire at 00.00am on this date. If we do not hear from you within this time then it will be assumed that the paper has been withdrawn. In exceptional circumstances, extensions may be possible if agreed with the Editorial Office in advance. We do not allow multiple rounds of revision so we urge you to make every effort to fully address all of the comments at this stage. If deemed necessary by the Editors, your manuscript will be sent back to one or more of the original reviewers for assessment. If the original reviewers are not available we may invite new reviewers.

Please also include the following statements alongside the other end statements. As we cannot publish your manuscript without these end statements included, if you feel that a given heading is not relevant to your paper, please nevertheless include the heading and explicitly state that it is not relevant to your work.

- Ethics statement

Please clarify whether you received ethical approval from a local ethics committee to carry out your study. If so please include details of this, including the name of the committee that gave consent in a Research Ethics section after your main text. Please also clarify whether you received informed consent for the participants to participate in the study and state this in your Research Ethics section.

OR

Please clarify whether you obtained the necessary licences and approvals from your institutional animal ethics committee before conducting your research. Please provide details of these licences and approvals in an Animal Ethics section after your main text.

OR

Please clarify whether you obtained the appropriate permissions and licences to conduct the fieldwork detailed in your study. Please provide details of these in your methods section.

- Data accessibility

It is a condition of publication that you make available the data and research materials supporting the results in the article. Datasets should be deposited in an appropriate publicly available repository and details of the associated accession number, link or DOI to the datasets must be included in the Data Accessibility section of the article (<https://royalsocietypublishing.org/rsos/for-authors#question17>). Reference(s) to datasets should also be included in the reference list of the article with DOIs (where available).

Please include a Data Availability section after your main text stating where supporting data are available from, or where they will be made available should your article be accepted for publication.

<http://datadryad.org/submit?journalID=RSOS&manu=RSOS-211223>

- Competing interests

Please include a Competing Interests section after your main text declaring any financial or non-financial competing interests. If you have no competing interests please state 'I/we have no competing interests.'

- Authors' contributions

Please include an Authors' Contributions section at the end of your main text detailing the contribution of each author. All authors should have read and approved the manuscript before submission and this should be stated in the Authors' Contributions section.

The list of Authors should meet all of the following criteria; 1) substantial contributions to conception and design, or acquisition of data, or analysis and interpretation of data; 2) drafting the article or revising it critically for important intellectual content; and 3) final approval of the version to be published.

- Acknowledgements

- Funding statement

Please include a funding section after your main text which lists the source of funding for each author.

Yours sincerely,
Dr Ellis Wilde
Publishing Editor, Journals

On behalf of the Subject Editor Professor Anthony Stace and the Associate Editor Dr Nadia Martinez Villegas.

RSC Associate Editor

Comments to the Author:

The research presented in this draft might be of interest to the RSOS audience. However, the manuscript must be improved. Please read carefully each of the comments from the reviewers and address each of them carefully.

RSC Subject Editor

Comments to the Author:

(There are no comments.)

Reviewers' Comments to Author:

Reviewer: 1

Comments to the Author(s)

Comments

The research article "Synthesis of a new chitosan-p-tert-butylcalix[4]arene polymer as adsorbent for toxic Hg²⁺ metal ion" seems to be a good work by the researchers, however it requires a major revision so as to be published in this reputed journal.

1. The introduction section needs improvement in terms of language and content. The authors should consult native English speaker and rewrite the introduction and rewrite the introduction with the updated references.

a. RSC Advances 2019, 10 (5), 2943-2943

b. ACS omega 2019, 4 (17), 17425-17437

c. International journal of biological macromolecules, 2019, 136, 189-198

2. The authors have focussed mainly on synthesis and characterization of the product, however the adsorption properties which is the application part needs attention.

3. Effect of pH on the adsorption sites and the adsorption of Hg²⁺ on chitosan-p-tert-butylcalix[4]arene polymer must be explained with point of zero charge of the adsorbent.

4. The selective adsorption of Hg²⁺ on chitosan-p-tert-butylcalix[4]arene polymer must be justified by adsorption kinetics and thermodynamic data. Furthermore, the spontaneity of the adsorption of Hg²⁺ on chitosan-p-tert-butylcalix[4]arene polymer must be explained using Van't Hoff plots

5. Error bars must be incorporate with all the data plots, to justify the selective adsorption of of Hg²⁺ on chitosan-p-tert-butylcalix[4]arene polymer.

6. Mere calculation of adsorption data from the kinetic and isotherm plots is not enough for conclusion of the adsorption phenomena. The validation of the results must be ensured by statistical error analysis by the sum of the square of the error (SSE), sum of absolute error (SAE), chi square (χ^2), and standard deviation Δq (%), between the experimental data and calculated values.

Reviewer: 2

Comments to the Author(s)

I would like to thank to the authors for their detailed research about this topic. This manuscript employs the adsorption of Hg²⁺ ions from water by using functionalized chitosan. The authors compared the adsorption capacity of newly synthesized functionalized-chitosan with the ones without functional group. Then, they characterized the structural properties through FTIR, XRD and SEM. The Hg⁺ adsorption capacity of synthesized material was investigated by ICP-MS. The research is quite comprehensive and includes enough physical characterization techniques for the synthesized polymers. Also, the authors studied the adsorption kinetics by Freundlich and Langmuir isotherms. According to the presented results, they reported that 25 mg of synthesized polymer could adsorb 94% of metal ions from 25 mL of 125 nmol/ml aqueous Hg²⁺ solution. The research contributes to the current literature about the utilization of functionalized-chitosan for metal ion removal applications.

I would recommend that this manuscript is suitable for publication in 'Royal Society of Open Science' after the consideration of suggested points.

You can find my comments and suggestions below.

- The language of the manuscript should be reviewed. There are repetitive conjunctions in same sentences.
- The authors should add the details about pH during the adsorption and they should include a pH study that shows the adsorption of Hg²⁺ vs pH of solution.

- There are some other studies which use chitosan for the metal ion removal. They might be cited such as Biomacromolecules 2010, 11, 12, [3301-3308](tel:3301-3308); RSC Adv., 2013,3, [7828-7837](tel:7828-7837).
- In Figure 1, the peaks might be labelled to show the functional groups.
- In section 3.5, the paragraph starts with 'SEM also known as SEM microscopy'. I think, it's not a true definition because SEM abbreviation already includes 'microscopy' term.
- In Figure 3 and 4, there are (a), (b), (c) sections that identify the magnification of micrographs. The authors gave the scale bar dimensions in the caption but it seems like the size of chitosan. If they can give the magnification values, the caption will be more clear.
- In Figure 6, the standard deviation of chitosan was given horizontally. Is it true? I think standard deviations give the deviation from average adsorption (qe) value but it seems like the deviation from time.
- In Section 3.9, the last sentence says that 'This means there are two kind of adsorption process between different functional groups of the CCP and Hg+2.' What do you mean here? Why does it depend on the concentration? Could you give more details about it?
- The results are given as % sorption by using X mg of polymer from X mL of X mol solution. I think it is pretty confusing. The discussion about the adsorption results should be done as mg (ion) per g (polymer) (mg/g) - the units may change. It makes the result comparable with the current literature.
- In the last part, there is a need for discussion about the results. The literature studies about the Hg+2 removal by using chitosan should be compared with this research's results. Why these polymers are promising and why people should prefer this method?
- What about the performance of polymers in the presence of competitive double valent ions? Do the authors have any result about selectivity of polymer?
- What is the reusability property of polymers? Can we regenerate them and how is the adsorption / desorption cycle?

Reviewer: 3

Comments to the Author(s)

Equation 2 is missing t, please amend.

Author's Response to Decision Letter for (RSOS-211223.R0)

See Appendix A.

RSOS-211223.R1 (Revision)

Review form: Reviewer 2 (Tugba Isik)

Is the manuscript scientifically sound in its present form?

Yes

Are the interpretations and conclusions justified by the results?

Yes

Is the language acceptable?

Yes

Do you have any ethical concerns with this paper?

No

Have you any concerns about statistical analyses in this paper?

No

Recommendation?

Accept as is

Comments to the Author(s)

I would like to thank to the authors for their clear and point by point response to the reviewer's comments. I would recommend that this manuscript is suitable for publication in 'Royal Society of Open Science'.

Decision letter (RSOS-211223.R1)

Dear Dr zadmard:

Title: Synthesis of a new chitosan-p-tert-butylcalix[4]arene polymer as adsorbent for toxic Hg²⁺ metal ion

Manuscript ID: RSOS-211223.R1

It is a pleasure to accept your manuscript in its current form for publication in Royal Society Open Science. The chemistry content of Royal Society Open Science is published in collaboration with the Royal Society of Chemistry.

Yours sincerely,

Kate Jones

Assistant Editor, Journals

Royal Society of Chemistry
Thomas Graham House
Science Park, Milton Road
Cambridge, CB4 0WF

Royal Society Open Science - Chemistry Editorial Office

On behalf of the Subject Editor Professor Anthony Stace and the Associate Editor Dr Nadia Martinez Villegas.

RSC Associate Editor
Comments to the Author:
(There are no comments.)

RSC Subject Editor
Comments to the Author:
(There are no comments.)

Reviewer(s)' Comments to Author:

Reviewer: 2

Comments to the Author(s)

I would like to thank to the authors for their clear and point by point response to the reviewer's comments. I would recommend that this manuscript is suitable for publication in 'Royal Society of Open Science'.

Appendix A

Prof. Dr. Reza Zadmard
Chemistry and Chemical Engineering
Research center of Iran, P.O. Box: 14335-186,
Tehran/ IRAN
Tel.: 0098-21-44787719
Fax: 0098-21-44787720
E-mail: Zadmard@ccerci.ac.ir

Dr. Ellis Wilde
Publishing Editor, Journals
Royal Society of Chemistry
Thomas Graham House
Science Park, Milton Road
Cambridge, CB4 0WF
Royal Society Open Science - Chemistry Editorial Office

Tehran, Jan 30, 2022

Dear Dr. Wilde,

Thank you for your e-mail dated 14 Jan. 2022 about our manuscript entitled: "*Synthesis of a new chitosan-p-tert-butylcalix[4]arene polymer as adsorbent for toxic Hg²⁺ metal ion*" (Manuscript ID: RSOS-211223). We thank the reviewers for their interest in our work and for helpful comments that will greatly improve the manuscript and we have tried to do our best to respond to the points raised. The Referees have brought up some good points and we appreciate the opportunity to clarify our research objectives and results.

As indicated below, we have checked all the comments provided by the reviewers and have made necessary changes accordingly to their indications. In the following answers, all reviewers' comments are in black, and our answers are in red. Please note that in the revised manuscript, the revised sentences are blue and the added sentences are red.

Response to Reviewer 1 Comments:

1. The introduction section needs improvement in terms of language and content. The authors should consult native English speaker and rewrite the introduction and rewrite the introduction with the updated references.

a. RSC Advances 2019, 10 (5), 2943-2943

b. ACS omega 2019, 4 (17), 17425-17437

c. International journal of biological macromolecules, 2019, 136, 189-198

Response:

According to the respected reviewer, these above mentioned 3 references (Ref. No. [34], [37] and [38]) were added to the introduction and the text was rewritten.

2. The authors have focused mainly on synthesis and characterization of the product, however the adsorption properties which is the application part needs attention.

Response:

The main purpose of this study was to synthesize and identify a new polymer based on calix[4]arene. Although we have provided some of the absorption properties of this polymer, our team is conducting additional studies for the application of this new polymer, the results of which will be presented in future publications.

3. Effect of pH on the adsorption sites and the adsorption of Hg^{2+} on chitosan-p-tert-butylcalix[4]arene polymer must be explained with point of zero charge of the adsorbent.

Response:

In this article, all studies have been done in comparison with chitosan polymer and chitosan is sensitive to pH and dissolves at acidic pH. Therefore, the adsorption process was studied at neutral pH (please see the reference 55). In addition, in some similar studies of our work, chitosan has been used as an adsorbent at a specific pH without investigating the effect of different pH (please see the reference 22).

4. The selective adsorption of Hg^{2+} on chitosan-p-tert-butylcalix[4]arene polymer must be justified by adsorption kinetics and thermodynamic data. Furthermore, the spontaneity of the adsorption of Hg^{2+} on chitosan-p-tert-butylcalix[4]arene polymer must be explained using Van't Hoff plots.

Response:

Thanks for your thoughtful comment. In this study, the adsorption process was investigated at ambient temperature. Therefore, it is not possible to study the absorption process using van't Hoff equation. There is also no claim that mercury absorption is specific in this study,

and it is possible that other metal cations also have interactions with this adsorbent, which will be investigated in a follow-up study.

5. Error bars must be incorporate with all the data plots, to justify the selective adsorption of Hg^{2+} on chitosan-p-tert-butylcalix[4]arene polymer.

Response:

It was corrected according to your comment.

6. Mere calculation of adsorption data from the kinetic and isotherm plots is not enough for conclusion of the adsorption phenomena. The validation of the results must be ensured by statistical error analysis by the sum of the square of the error (SSE), sum of absolute error (SAE), chi square (χ^2), and standard deviation Δq (%), between the experimental data and calculated values.

Response:

This is a reasonable suggestion which is very much appreciated. We revised accordingly to include this information. We calculated all requested values and summarized in the table below.

No.	Error Functions	Langmuir	Freundlich	pseudo first-order kinetic	pseudo second-order kinetic
1	The Sum Square Error (SSE)	1.44592	0.9249	0.2522	0.0768
2	The Average Relative Error (ARE)	0.2825	0.2421	0.2151	0.0628
3	The Sum of Absolute Error (SAE)	1.99	1.4571	0.45	0.06
4	Non-linear Chi-Square Test (Chi-Sq/ χ^2)	1.0556	0.5133	0.1313	0.0085
5	Residual sum of squares (SSR)	0.1065	0.4246	0.0003	0.0005

The following Table has been added in the updated version; (3.8 Adsorption kinetic, page 15)

	Pseudo-first order	Pseudo-second order
k_i	-0.0287	0.0086
q_e	57.0164	66.6666
ARE^*	0.2151	0.0628
R^2	0.9497	0.9932

Furthermore, below sentence (highlighted in green) has been added in the manuscript (3.10 Adsorption isotherm section, page 17), *as can be seen in the figure 11, R^2 value is 0.94, which means that the experimental information conformed to the Freundlich model.* Moreover, the Average Relative Error (ARE) for Freundlich isotherm model is calculated as 0.2421 which is significantly less than that for Langmuir isotherm model (0.2825).

Response to Reviewer 2 Comments:

The language of the manuscript should be reviewed. There are repetitive conjunctions in same sentences.

Response:

The text was rewritten and the corrections were made according to your comment as much as possible.

The authors should add the details about pH during the adsorption and they should include a pH study that shows the adsorption of Hg^{2+} vs pH of solution.

Response:

Thanks for your comment. As was mentioned in response to reviewer 1 (comment 3), In this article, all studies have been done in comparison with chitosan polymer which is sensitive to pH and dissolves at acidic pH. Therefore, the adsorption process was studied at neutral pH (please see the reference 55). In addition, in some similar studies of our work, chitosan has been used as an adsorbent at a specific pH without investigating the effect of different pH (please see the reference 22).

There are some other studies which use chitosan for the metal ion removal. They might be cited such as Biomacromolecules 2010, 11, 12, 3301–3308; RSC Adv., 2013,3, 7828-7837.

Response:

These 2 references (Ref. No. [22] and [36]) were added to the introduction and the text was rewritten.

In Figure 1, the peaks might be labelled to show the functional groups.

Response:

Figure 1 was modified according to your suggestion. (in the manuscript, page 9)

In section 3.5, the paragraph starts with ‘SEM also known as SEM microscopy’. I think, it’s not a true definition because SEM abbreviation already includes ‘microscopy’ term.

Response:

It was corrected according to your comment. (in the manuscript, page 11)

In Figure 3 and 4, there are (a), (b), (c) sections that identify the magnification of micrographs. The authors gave the scale bar dimensions in the caption but it seems like the size of chitosan. If they can give the magnification values, the caption will be more clear.

Response:

Figures 3 and 4 were modified according to your comment. (in the manuscript, page 11)

In Figure 6, the standard deviation of chitosan was given horizontally. Is it true? I think standard deviations give the deviation from average adsorption (q_e) value but it seems like the deviation from time.

Response:

Figure 6 was corrected according to your comment. (in the manuscript, page 13)

In Section 3.9, the last sentence says that ‘This means there are two kind of adsorption process between different functional groups of the CCP and Hg^{+2} .’ What do you mean here? Why does it depend on the concentration? Could you give more details about it?

Response:

Since the polymer adsorbent consists of different functional groups as hydroxyl group, amid bond, aromatic rings for π interactions and etc., here (in the manuscript, section 3.9, page 15) the authors aim to show that, according to the diagram, different interactions are involved in the adsorption process but apparently misleads the reader. Therefore, this sentence was deleted.

The results are given as % sorption by using X mg of polymer from X mL of X mol solution. I think it is pretty confusing. The discussion about the adsorption results should be done as

mg (ion) per g (polymer) (mg/g) – the units may change. It makes the result comparable with the current literature.

Response:

Thank you for your accurate comment. Your opinion is quite correct. Most studies in this field report the adsorption capacity in (mg/g), but there are a number of articles that report the adsorption capacity as a percentage of adsorption, from which we took a model. A comparison table and relevant references were added in Section 3.6 (in the manuscript, page 12), and the confusing sections were rewritten.

In the last part, there is a need for discussion about the results. The literature studies about the Hg⁺² removal by using chitosan should be compared with this research's results. Why these polymers are promising and why people should prefer this method?

Response:

A comparison table was added in Section 3.6 of the text (page 12). The high efficiency in the synthesis and powderiness of this new compound improves its use as an adsorbent. Also, the resistance of this polymer in acidic environments and its insolubility in a wide range of pH are other advantages of this new synthetic polymer.

What about the performance of polymers in the presence of competitive double valent ions? Do the authors have any result about selectivity of polymer?

Response:

The main purpose of this study was to focus on the synthesis and identification of a new polymer, and currently, our team is conducting additional studies for the application of this new polymer, including the issues of selectivity and competitive adsorption in the presence of other ions.

What is the reusability property of polymers? Can we regenerate them and how is the adsorption / desorption cycle?

Response:

Thank you for your comments. The desorption process was not studied in this work. A search of the literature shows that there are many articles that, like our work, have not studied desorption despite studying the absorption process (please see the references 21, 22 and 55).

Response to Reviewer 3 Comments:

1. Equation 2 is missing t, please amend.

Response:

It was corrected. (in the manuscript, page 6)

I'm looking forward to hearing from you in near future.

Sincerely,

Reza Zadmard